# Evaluation of the Humoral Immune Response of a Heterologous Vaccination between BBIBP-CorV and BNT162b2 with a Temporal Separation of 7 Months, in Peruvian Healthcare Workers with and without a History of SARS-CoV-2 Infection

**DOI:** 10.3390/vaccines10040502

**Published:** 2022-03-24

**Authors:** Miguel Hueda-Zavaleta, Juan C. Gómez de la Torre, José Alonso Cáceres-Del Aguila, Cecilia Muro-Rojo, Nathalia De La Cruz-Escurra, Daniella Arenas Siles, Diana Minchón-Vizconde, Cesar Copaja-Corzo, Fabrizzio Bardales-Silva, Vicente A. Benites-Zapata, Alfonso J. Rodriguez-Morales

**Affiliations:** 1Faculty of Health Sciences, Universidad Privada de Tacna, Tacna 23003, Peru; diaminchon@virtual.upt.pe (D.M.-V.); cescopajac@upt.pe (C.C.-C.); 2Hospital III Daniel Alcides Carrión-Essalud Tacna, Tacna 23000, Peru; 061210@colegiomedico.org.pe; 3Laboratorio Clínico Roe, Lima 15076, Peru; jgomez@labroe.com (J.C.G.d.l.T.); jcaceres@labroe.com (J.A.C.-D.A.); cmuro@labroe.com (C.M.-R.); nathalia.delacruz@labroe.com (N.D.L.C.-E.); 4Faculty of Medicine, Universidad Científica del Sur, Lima 15067, Peru; darenassi@cientifica.edu.pe; 5Hospital Hipólito Unanue—Tacna, Tacna 23003, Peru; 6Unidad de Investigación para la Generación y Síntesis de Evidencias en Salud, Universidad San Ignacio de Loyola, Lima 15024, Peru; vbenites@usil.edu.pe; 7Master Program on Clinical Epidemiology and Biostatistics, Universidad Científica del Sur, Lima 15046, Peru; arodriguezmo@cientifica.edu.pe; 8Grupo de Investigación Biomedicina, Faculty of Medicine, Fundación Universitaria Autónoma de las Américas, Pereira 660003, Risaralda, Colombia

**Keywords:** COVID-19 vaccines, SARS-CoV-2, humoral immunity, heterologous booster vaccination, neutralizing antibodies

## Abstract

Information on the effects of a heterologous booster in adult patients first vaccinated with the BBIBP-CorV vaccine is limited. This prospective cohort study evaluated the humoral response of 152 healthcare workers (HCWs) from a private laboratory in Lima (Peru) before and after receiving the BNT162b2 vaccine, with a seven-month interval since the BBIBP-CorV doses. We employed the Elecsys^®^ anti-SARS-CoV-2 S and the cPass™ SARS-CoV-2 Neutralization Antibody (NAbs) assays to evaluate anti-S-RBD IgG and NAbs, respectively. Of the 152 HCWs, 79 (51.98%) were previously infected (PI) with SARS-CoV-2 and 73 (48.02%) were not previously infected (NPI). The proportion of HCWs with positive NAbs, seven months after the BBIBP-CorV immunization, was 49.31% in NPI and 92.40% in PI. After the booster, this ratio increased to 100% in both groups. The anti-S-RBD IgG and NAbs in the HCWs’ NPI increased by 32.7 and 3.95 times more, respectively. In HCWs’ PI, this increment was 5 and 1.42 times more, respectively. There was no statistical association between the history of previous SARS-CoV-2 infection and the titer of anti-S-RBD IgG and NAbs after the booster. The humoral immunity presented a robust increase after receiving the BNT162b2 booster and was more pronounced in NPI.

## 1. Introduction

Nowadays, nine vaccines against the disease caused by the severe acute respiratory syndrome coronavirus 2 virus (SARS-CoV-2) have been demonstrated to be effective and have been authorized for emergency use by the World Health Organization (WHO) [1]. These vaccines have substantially reduced the critical cases and deaths provoked by COVID-19 [2]. However, their covering time is still uncertain and could vary between vaccine formulations and populations. In Peru, since February 2021, all healthcare workers (HCW) have received two doses of the inactivated virus vaccine against the SARS-CoV-2, BBIBP-CorV (Sinopharm), which has demonstrated a good immunogenic profile, safety, and efficacy on clinical trials [3]. The neutralizing antibodies (NAbs), induced by the vaccine against the COVID-19, can block the entry of the SARS-CoV-2 into human cells [4]. There is an inverse correlation between their presence and the risk of getting this infectious disease [5]. Nevertheless, a decline has been reported with respect to NABs levels around the third to sixth months after natural infection with SARS-CoV-2 [6], as well as in patients immunized by BNT162b2 (Pfizer-BioNTech, Chesterfield, MO, USA) and ChadOx1 (Oxford-AstraZeneca, Oxford, United Kingdom), and mainly on those older than 65 years and immunosuppressed [7,8,9]. This decline coincides with the reduction of the vaccine’s effectiveness across time [10]. This reduction of the efficacy over time and the raising of SARS-Cov2 variants [8,11,12] has motivated many countries to approve a booster vaccine dose (homologous or heterologous) against COVID-19, after a not standardized period, in previously vaccinated patients. On 15 October 2021, the Minister of Health in Peru started a booster vaccination campaign against the COVID-19 on their healthcare workers (HCW) [13].

To date, there is limited evidence on the effectiveness of a heterologous BNT162b2 booster vaccine in patients who have previously received two protocol doses of BBIBP-CorV. Our study evaluated humoral response quantifiably, employing the Elecsys^®^ anti-SARS-CoV-2 S assay and the cPass™ SARS-CoV-2 Neutralization Antibody Detection Kit before and after the vaccination with BNT162b2 in healthcare workers immunized seven months previously with BBIBP-CorV (according to the history of previous SARS-CoV-2 infection and or seropositivity).

## 2. Materials and Methods

The design of the study was a prospective cohort in a total of 152 HCWs from a private laboratory in Lima, Peru, who have received a heterologous booster vaccination with the COVID-19 mRNA vaccine BNT162b1 (BioNTech SE, Mainz, Germany) (Pfizer, New York, NY, USA), seven months after the second dose of the inactivated vaccines against SARS-CoV-2, the BBIBP-CorV (Sinopharm, Shanghai, China). The period of the study was between February 2021 and January 2022. The study protocol was approved by the institutional bioethics committee (CIB) “Vía Libre” (IRB: 8115) on 30 November 2021. All participants gave their authorization to participate in the study through an informed consent that they signed.

Two groups of participants have evaluated before the NAbs dosage at seven months after the second dose with BBIBP-CorV: (1) a group with evidence of previous SARS-CoV-2 infection (PI) and (2) a group without previous SARS-CoV-2 infection (NPI). To discern between the HCWs PI and NPI, we consider any molecular test (qRT-PCR) positive results from the HCWs before and after receiving vaccine BBIBP-CorV and their historical serology records before receiving their first dose with BBIBP-CorV-2. Historical serology was determined by Ab against the N antigen, using the Elecsys^®^ Anti-SARS-CoV-2 (Roche Diagnostics International AG, Rotkreuz, Switzerland) and Ab against the S1 antigen, employing the Aeskulisa SARS-CoV-2 S1 IgG and IgM (Aesku. Di-agnostics GmbH & Co. KG, Wendelsheim, Germany).

Once the participants were typified, blood samples were collected from them in two periods: (1) seven months after the second dose with BBIBP-CorV and (2) 21 days after de heterological booster with BNT162b2. For evaluating the presence of NAbs against SARS-CoV-2, we employed the cPass™ SARS-CoV-2 Neutralization Antibody Detection Kit (GenScript, Piscataway, NJ, USA), also known as SARS-CoV-2 surrogate virus neutralization test (sVNT). It is a blocking enzyme-linked immunosorbent assay (ELISA) intended for the qualitative and semi-quantitative direct detection of total neutralizing antibodies to SARS-CoV-2 based on mixing the patient’s serum with receptor-binding domain (RBD) conjugated with recombinant horseradish peroxidase (HRP) and the angiotensin-converting enzyme 2 (ACE2). If the patients have developed NAbs, the interaction of RBD-ACE2 would be interrupted, provoking a signal lost from the HRP. At the same time, if there is an absence of NAbs in the sample, the HRP-RBD would bind with the ACE2 and generate a colorimetric signal. The inhibition signal percentage (PSI) was determined by subtracting one minus the division of the sample’s optical densities (OD) with the OD of the negative control, multiplied by 100. A PSI bigger or equal to 30% was considered positive [14]. We also employed the Elecsys^®^ anti-SARS-CoV-2 S test (Roche Diagnostics International AG, Rotkreuz, Switzerland). This quantitative immunoassay detects high-affinity total Abs (predominantly IgG) against the S protein’s receptor-binding domain (RBD) from the SARS-CoV-2. A value higher or equal than 0.8 U/mL was considered positive [15].

Statistical analysis was performed in STATA V17.0 software (StataCorp., College Station, TX, USA) and Prism V 9.2.0 (Graphpad Software, LLC, San Diego, CA, USA). The categorical variables were presented as absolute and relative frequency, the numerical variables were reported as the median and interquartile range (IQR). As suggested by the WHO guidelines on clinical evaluation of vaccines, antibody titers were reported as geometric means (GMT) with their 95% confidence interval (95% CI) [16]. The bivariate analysis of the categorical variables was analyzed using the chi-square test (χ^2^) or Fisher’s exact test, depending on their expected values. We used the Mann-Whitney U statistical test to compare the quantitative variables since its distribution was not symmetrical. The comparison between NAbs before and after the booster in each group was determined through the non-parametric statistical test-Wilcoxon of signs and ranges. A value of *p* < 0.05 was considered statistically significant. To evaluate the factors associated with higher levels of NAbs and anti-RBD antibodies, crude and adjusted linear regression models were used, calculating the β values and their respective 95% IC. Only those variables that met the assumptions of linearity, independence of the observations, normality of the residuals, and homoscedasticity were included in the analysis.

## 3. Results

From all of the HCWs immunized with heterologous booster vaccine BNT162b2, seven months after the second dose of the BBIBP-CorV, 152 were included in the study, 73 (48.02%) NPI and 79 (51.98%) PI. Of all of the PI patients, 62 acquired COVID-19 before the first immunization with BBIBP-CorV. The remaining 17 became infected after being vaccinated with BBIBP-CorV. The median age was 34 years, and 119 (78.3%) were females. Statistically, no significant differences were observed in any of these variables between PI and NPI.

### 3.1. Elecsys^®^ Anti-SARS-CoV-2 S (Anti-S-RBD IgG)

The proportion of HCWs with positive values of anti-S-RBD IgG seven months after the second dose with BBIBP-CorV was 99.3%, being 100% on PI and 98.6% in those NPI (*p* = 0.480). This positive proportion increased after the heterologous booster vaccine, on which 100% of NPI were positive, and the PI group remained similarly high (Table 1).

The level of Ab anti-S-RBD IgG increased significantly after the heterologous booster with BNT162b2 on NPI HCWs, GMT titer of 69.02 (95% CI: 44.91 to 106.07) to a GMT titer of 2260 (95% CI: 2068.72 to 2470.09). This represents an increase in the titers of anti-S-RBD IgG after the booster BNT162b2 of 32.7 more on those NPI (Table 1, Figure 1a).

### 3.2. cPass™ SARS-CoV-2 Neutralization Antibody

At seven months after the second dose with BBIBP-CorV, we observed that the proportion of patients with positive NAbs was 49.3% on NPI and 92.40% on PI (*p* < 0.001). This positivity increased after receiving the BNT162b2 booster on each group to 100% (Table 1).

An increase of the PSI was observed on the HCWs NPI, from a GMT of 23.89 (95% CI: 18.70–30.50) to 94.54 (95% CI: 92.64–96.48). On the HCWs PI group, we observed a GMT increase from 67.07 (95% CI: 58.89 to 76.38) to 95.40 (95% CI: 93.46 to 97.38). This finding reflects that the heterologous booster generates an increase in the PSI evaluated with cPass™ of 3.95 and 1.42 times more on HCWs NPI and PI, respectively (Table 1, Figure 1b).

Finally, we determined employing linear regression that the HCWs PI showed 0.859 points of PSI higher of NAbs with cPass™ (*p* = 0.414), and 54.002 IU/mL more of titers with Elecsys^®^ anti-SARS-CoV-2 S (*p* = 0.420) than those HCWs NPI. However, this particularity was not statistically significant. We also observed that male participants showed higher antibody titers than women, but this was not statistically significant either (Table 2).

## 4. Discussion

The present cohort study evaluated HCWs that received a heterologous booster with mRNA vaccine BNT162b2 after two doses with the inactivated virus vaccine BBIBP-CorV. We observed that only half of the HCWs NPI showed positive NAbs results, seven months after the second dose with BBIBP-CorV. Nonetheless, this proportion increased substantially after the vaccination with the heterologous booster to 100% on HCWs, PI, and NPI. The NAbs and anti-S-RBD IgG levels increased robustly after the booster administration closer to the maximum detection level in both studied groups. Also, the levels of NAbs and anti-S-RBD IgG reached were not influenced by any background of the previous infection or not, an aspect that was evident before getting the booster.

There is evidence suggesting that a three-dose (homologous or heterologous) vaccination schedule against COVID-19 is superior to just two doses; the COV-BOOST clinical trial evaluated the safety and immunogenicity of seven vaccines against COVID-19 as a third dose (AstraZeneca, Curevac, Janssen, Moderna, Novavax, Pfizer, and Valneva) after 70 days or more of receiving a two-dose schedule of ChAdOx1-S/nCoV-19 (AstraZeneca) or BNT-162b2 (BioNTech, Pfizer). All of the vaccines tested showed acceptable side effect profiles and reactogenicity compared to the control group. The Moderna vaccine achieved the highest immunogenicity [17]. These results would indicate that immunogenicity could be higher when immunization is heterologous and with mRNA-based vaccines [18].

The effectiveness of a BNT162b2 booster, after five months at least in patients with two doses of the same type of vaccine, was 90% of death prevention for COVID-19 compared to those who did not receive this booster [19]. Similarly, another study demonstrated a booster with BNT162b2 six months after the second dose with the same vaccine, confers protection of 89% against COVID-19 infection [20]. However, these conclusions were before the emergence of the B.1.529 variant (omicron), which (as a result of the numerous mutations on the S protein and its capacity to evade the NAbs) has generated a decrease in the vaccine’s effectiveness. A pre-print study in England determined that the effectiveness of preventing the symptomatic infection provoked by the omicron variant after two doses with BNT162b2 was only 30%, and after a homologous booster, this effectiveness was 75% [21]

We observed a significant decrease in NAbs in patients vaccinated with BBIBP-CorV over time, consistent with other studies [22,23,24]. Lui Y. et al., report that the median concentration of NAbs after administering a BBIBP-CorV homologous boost increased up to 10-fold [24]. In contrast to Moghnieh R. et al., who reported an increase in NAbs of more than 800-fold after a heterologous BNT162b2 boost in patients previously vaccinated with two doses of BBIBP-CorV [23].

Similar findings have been reported in patients who received a homologous booster or heterologous after two doses of the vaccine CoronaVac (Sinovac Biotech, Beijing, China). A randomized clinical trial evaluated the humoral response after a booster with the cAd26.COV2-S (Janssen), BNT162b2 (Pfizer-BioNTech), or AZD1222 (AstraZeneca) vaccine, compared with the third dose of CoronaVac in patients who had received two doses of CoronaVac six months earlier. Observing that the four vaccines evaluated caused a significant increase in NAbs and booster BNT162b2 was the most immunogenic [25]. These findings are consistent with reports from other observational studies that assessed the effect of BNT162b2 and AZD1222 as a booster in previously vaccinated by CoronaVac, providing a marked increase in humoral response compared to a homologous booster [26,27,28]. However, the IgG anti-S-Rbd level with the AZD1222 heterologous reinforcement was lower than the observed in our study with BNT162b2 (1.4 times lower).

Nevertheless, even after a BNT162b2 booster in previously immunized with CoronaVac, a reduction of the neutralizing activity against the omicron variant of six times more than the wild type was observed. At the same time, those who only had two doses did not have a presence of NAbs against the omicron variant [29].

Currently, few studies have evaluated the immune response to heterologous BNT162b2 boosting after two doses of BBIBP-CorV. Our study had some limitations. The majority of participants were young women without comorbidities, which could affect the extrapolation of the results in the general population. Our findings focused on the characterization of the humoral response, not the cellular immunity mediated response, which seems to have a fundamental role in preventing severe cases of COVID-19. Likewise, a post-booster clinical follow-up was not conducted to investigate its effectiveness in preventing COVID-19, nor the adverse effects of the booster. Finally, a control group was not available as a BBIBP-CorV homologous booster since, in Peru, only BNT162b2 was approved as a booster in HCW.

## 5. Conclusions

A heterologous booster of the BNT162b2 mRNA vaccine, after seven months of two doses of the inactivated vaccine against SARS-CoV-2 BBIBP-CorV, induced a robust immune response, regardless of the history of previous infection or not.

## Figures and Tables

**Figure 1 vaccines-10-00502-f001:**
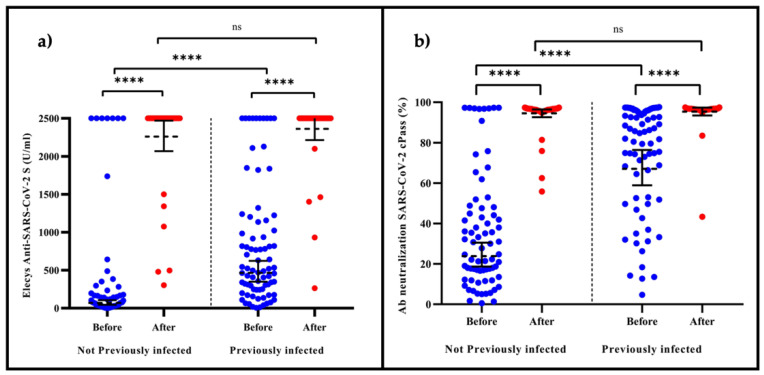
(**a**) The effect at 21 days of the heterologous booster with BNT162b2 evaluated with Elecsys^®^ anti-SARS-CoV-2 S. The boxplot shows the GMT titers with 95% CI of the anti-S-RBD antibodies determined by Elecsys^®^ Anti-SARS-CoV-2 S. They were significantly higher after applying the heterologous booster of the BNT162b2, in HCW-PI (*p* < 0.001) ^†^ and NPI (*p* < 0.001) ^†^. Higher titers were observed before receiving the booster in PI patients than NPI (*p* < 0.001) ^††^. However, no differences in anti-S-RBD GMTs were observed after receiving booster BNT162b2 between HCW PI and NPI (*p* = 0.340) ^†^. (**b**) The effect at 21 days of the heterologous booster with BNT162b2 evaluated with cPass™ SARS-CoV-2 neutralization antibody detection. The boxplot shows the GMT titers with 95% CI of the NAbs determined by cPass™ (%). Those were significantly higher after applying the heterologous booster of the BNT162b2 in HCW PI (*p* < 0.001) ^†^ and NPI (*p* < 0.001) ^†^. Even when higher titers were observed on PI than NPI (*p* < 0.001) ^††^ before receiving this booster, no differences in NAb GMTs were observed after receiving booster BNT162b2 between HCW PI and NPI (*p* = 0.520) ^††^. ns, not significant; **** *p* < 0.0001; ^†^ Wilcoxon sign rank statistical test; ^††^ U Man-Whitney test; GMT, geometric mean; 95% CI, 95% confidence intervals; Nabs, neutralizing antibodies; HCW, health workers; PI, previously infected; NPT, not previously infected.

**Table 1 vaccines-10-00502-t001:** Demographic characteristics, humoral response rates by SARS-CoV-2 specific antibody levels before and after BNT162b2 booster of the study population and comparison between previously infected and previously uninfected.

Variable	Total (*n* = 152)	Previously Infected (*n* = 79)	Not Previously Infected (*n* = 73)	*p*-Value
Age, years * (IQR)	34.0 (28–42)	34 (27–40)	34 (28–43)	0.455 ^a^
Sex (%)				0.952 ^b^
- Female	119 (78.3)	62 (52.10)	57 (47.90)	
- Male	33 (21.7)	17 (51.52)	16 (48.48)	
Humoral response rates				
7 months after second dose				
- Ab neutralization cPass (%)	109 (71.7)	73 (92.4)	36 (49.3)	<0.001 ^c^
- Elecsys^®^ Anti-SARS-CoV-2 S (%)	151 (99.3)	79 (100)	72 (98.6)	0.480 ^c^
- Ab neutralization cPass (%) (95% IC) **	40.85 (34.90–47.81)	67.07 (58.89–76.38)	23.89 (18.70–30.50)	<0.001 ^a^
- Elecsys^®^ Anti-SARS-CoV-2 S (U/mL) (95% IC) **	186.26 (138.63–250.25)	466.11 (349.02–622.49)	69.02 (44.91–106.07)	<0.001 ^a^
21 days after booster				
- Ab neutralization cPass (%)	152 (100)	79 (100)	73 (100)	0.999 ^c^
- Elecsys^®^ Anti-SARS-CoV-2 S (%)	152 (100)	79 (100)	73 (100)	0.999 ^c^
- Ab neutralization SARS-CoV-2 cPass (%) (95% IC) **	94.99 (93.63–96.36)	95.40 (93.46–97.38)	94.54 (92.64–96.48)	0.520 ^a^
- Elecsys^®^ Anti-SARS-CoV-2 S (U/mL) (95% IC) **	2312.03 (2191.30–2439.42)	2360.69 (2213.49–2517.67)	2260 (2068.72–2470.09)	0.340 ^a^

* Median and interquartile range ** Geometric means and 95% confidence interval, ^a^ U-Mann Whitney test, ^b^ χ^2^, ^c^ Fisher’s exact test. IQR, interquartile range; 95% IC, 95% confidence interval; Ab, antibody.

**Table 2 vaccines-10-00502-t002:** Simple and multiple linear regression of the variables associated with titles of Ab neutralization SARS-CoV-2 cPass™, Elecsys^®^ anti-SARS-CoV-2 S-RBD after booster BNT162b2.

Ab Neutralization SARS-CoV-2 cPass™ after Booster BNT162b2				
**Variable**	**Crude β (95% IC)**	***p*-Value**	**Adjusted β (95% IC)**	***p*-Value**
Male	1.564 (−0.939 to 4.068)	0.219	1.569 (−0.937 to 4.076)	0.218
Previously infected	0.859 (−1.213 to 2.931)	0.414	0.865 (−1.203 to 2.934)	0.410
**Elecsys^®^ anti-SARS-CoV-2 S after Booster BNT162b2**				
**Variable**	**Crude β (95% IC)**	***p*-Value**	**Adjusted β (95% IC)**	***p*-Value**
Male	135.669 (−23.051 to 294.390)	0.093	135.989 (−22.914 to 924.892)	0.093
Previously infected	54.002 (−77.925 to 185.930)	0.420	54.544 (−76.581 to 185.670)	0.412

95% IC, 95% confidence interval; SARS-CoV-2, severe acute respiratory syndrome coronavirus 2.

## Data Availability

The data analyzed in this manuscript, as well as its definitions, could be downloaded from the following doi:10.17632/ztps23g4sv.2.

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
