# Peer review of "Evaluation of the Humoral Immune Response of a Heterologous Vaccination between BBIBP-CorV and BNT162b2 with a Temporal Separation of 7 Months, in Peruvian Healthcare Workers with and without a History of SARS-CoV-2 Infection"

_vaccines, 2022, doi:10.3390/vaccines10040502_

Round 1

Reviewer 1 Report

see attached

Reviewer 2 Report

This study examined the effect of a booster dose with BNT162b2 on health care workers in Peru who were previously vaccinated with two doses of the BBIBP-CorV inactivated vaccine. Two groups of HCWs were studied, viz. those who were previously infected with SARS-CoV-2 and those who were not. 

The manuscript was reasonably well written and supported by appropriate tables and figures, with a suitable bibliography. However, parts of the text were difficult to understand and would benefit from editing by a native English speaker (although I am most sympathetic towards the authors).

Unfortunately, the results of the study only confirm the prediction-that the BNT162b2 vaccine would stimulate the immune response-and as a result, the study will be of limited interest internationally.  The authors might consider a truncated version which presents the results in a more succinct manner.

Several points should be clarified;

Lines 76-81; the design of the study was difficult to understand and these lines should be re-written.

Lines 94 et seq; the design of the anti-RBD assay should be clarified.  Presumably this is a solid phase ELISA.

The term "third dose booster" is confusing and the authors should use "third dose" or "booster"

Table 1; I was confused as the figures in the table did not agree with the figures presented in the text.  In addition, the percentages in parentheses in the individual columns  should reflect the data in that column, so that (for example) 79 of 79 PI with anti-S Ab should be presented as 100%, not 52.32 as shown in the table. 

The terms IC and RIC should be defined.

Is it possible to adjust the assay to ensure that the RIC is not maximum 21 days after the booster?

Table 3 shows no data for female HCWs, although they represent a majority of individuals who were studied.

Perhaps the authors might consider reducing the length of the discussion to focus more on their data by moving the last paragraph to the start of the discussion, and to restrict the discussion to results of boosters with the BNT162b2 vaccine.

Author Response

Reviewer 1 and Responses:

  1. This study examined the effect of a booster dose with BNT162b2 on health care workers in Peru who were previously vaccinated with two doses of the BBIBP-CorV inactivated vaccine. Two groups of HCWs were studied, viz. those who were previously infected with SARS-CoV-2 and those who were not.
  • Indeed, that was the methodology.
  1. The manuscript was reasonably well written and supported by appropriate tables and figures, with a suitable bibliography. However, parts of the text were difficult to understand and would benefit from editing by a native English speaker (although I am most sympathetic towards the authors).
  • We appreciate the reviewer’s suggestion; we have revised and improved the narration of the manuscript.
  1. Unfortunately, the results of the study only confirm the prediction that the BNT162b2 vaccine would stimulate the immune response and as a result, the study will be of limited interest internationally. The authors might consider a truncated version which presents the results in a more succinct manner.
  • We appreciate the reviewer’s comment; we have reduced the length of the manuscript.
  1. Several points should be clarified; Lines 76-81; the design of the study was difficult to understand, and these lines should be re-written.
  • We appreciate the reviewer's observation; we have made the following modification (line 86-95):
  • Two groups of participants have evaluated previous to the NAbs dosage at seven months after the second dose with BBIBP-CorV: 1) a group with evidence of previous SARS-CoV-2 infection (PI) and 2) a group without previous SARS-CoV-2 infection (NPI). To discern between the HCWs PI and NPI, we consider any molecular test (qRT-PCR) positive results from the HCWs before and after receiving vaccine BBIBP-CorV and their historical serology records before receiving their first dose with BBIBP-CorV-2. Historical serology was determined by Ab against the N antigen, using the Elecsys® Anti-SARS-CoV-2 (Roche Diagnostics International AG, Rotkreuz, Switzerland) and Ab against the S1 antigen, employing the Aeskulisa SARS-CoV-2 S1 IgG and IgM (Aesku. Diagnostics GmbH & co. KG, Wendelsheim, Germany).
  1. Lines 94 et seq; the design of the anti-RBD assay should be clarified. Presumably this is a solid phase ELISA. The term "third dose booster" is confusing and the authors should use "third dose" or "booster"
  • We appreciate the comment of the reviewer, it is right to improve the explanation of the design, that is why we made the following modification (line 98-105): “For evaluating the presence of NAbs against SARS-CoV-2, we employed the cPass™ SARS-CoV-2 Neutralization Antibody Detection Kit (GenScript, Piscataway, NJ, EE.UU.), also known as SARS-CoV-2 surrogate virus neutralization test (sVNT). It is a Blocking Enzyme-Linked Immunosorbent Assay (ELISA) intended for the qualitative and semi-quantitative direct detection of total neutralizing antibodies to SARS-CoV-2 based on mixing the patient’s serum with receptor-binding domain (RBD) conjugated with recombinant horseradish peroxidase (HRP) and the angiotensin-converting enzyme 2 (ACE2).”
  • We also replaced the terms "third booster dose" with "booster" and "180 days" with "7 months" throughout the manuscript.
  1. Table 1; I was confused as the figures in the table did not agree with the figures presented in the text. In addition, the percentages in parentheses in the individual columns should reflect the data in that column, so that (for example) 79 of 79 PI with anti-S Ab should be presented as 100%, not 52.32 as shown in the table.
  • The reviewer is correct; we have revised the wording of the manuscript and corrected the details mentioned
  1. The terms IC and RIC should be defined.
  • The reviewer is right; we have added the definitions. RIC is IQR: Interquartile Range; IC is CI: Confidence Interval.
  1. Is it possible to adjust the assay to ensure that the RIC is not maximum 21 days after the booster?
  • We understand the reviewer's suggestion. Unfortunately, the evaluation time was taken as 21 days after the booster in the developed protocol.
  1. Table 3 shows no data for female HCWs, although they represent a majority of individuals who were studied.
  • Indeed, we did not include the information of women health workers since they were the point of comparison to find men’s results. But to improve the interpretation, we decided to add (line 177-178): "We also observed that male participants showed higher antibody titers compared to women, but this was not statistically significant either."
  1. Perhaps the authors might consider reducing the length of the discussion to focus more on their data by moving the last paragraph to the start of the discussion, and to restrict the discussion to results of boosters with the BNT162b2 vaccine.
  • We appreciate the reviewer's suggestion; we have followed his advice, generating a substantial discussion modification.

Reviewer 3 Report

The manuscript “Evaluation of the humoral immune response of a heterologous vaccination between BBIBP-CorV and BNT162b2 with a temporal separation of 7 months, in Peruvian healthcare workers with and without a history of SARS-CoV-2 infection” submitted by Hueda-Zavaleta M. et al. evaluates the humoral immunity elicited by heterologous combination of two different SARS-CoV-2 vaccine platforms, the BBIBP-CorV vaccine based on inactivated virus and the BNT162b2 vaccine based on mRNA. The authors analysed the longevity of humoral response (anti-RBD binding IgG antibodies and NAbs) induced by two BBIBP-CorV immunizations at 6 months after the last dose and the impact of a late boost with the heterologous BNT162b2 vaccine on the previous immunity. A robust increase of humoral immunity was detected after the heterologous mRNA boost, supporting previous findings.

The authors claimed that they could not find any study that evaluate the heterologous booster of BNT162b2 after two doses with BBIBP-CorV to highlight the novelty of the study, but several previous studies have addressed this evaluation:

-Rima Moghnieh, Rana Mekdashi, Salam El-Hassan, Dania Abdallah, Tamima Jisr, Mohammad Bader, Ihab Jizi, Mohamed H. Sayegh, Abdul Rahman Bizri, Immunogenicity and reactogenicity of BNT162b2 booster in BBIBP-CorV-vaccinated individuals compared with homologous BNT162b2 vaccination: Results of a pilot prospective cohort study from Lebanon. Vaccine,Volume 39, Issue 46,2021, Pages 6713-6719.

-Comparison of antibody and T cell responses elicited by BBIBP-CorV (Sinopharm) and BNT162b2 (Pfizer-BioNTech) vaccines against SARS-CoV-2 in healthy adult humans. Vályi-Nagy I, Matula Z, Gönczi M, Tasnády S, BekÅ‘ G, Réti M, Ajzner É, Uher F. Geroscience. 2021 Oct;43(5):2321-2331.

-Keskin AU, Bolukcu S, Ciragil P, Topkaya AE. SARS-CoV-2 specific antibody responses after third CoronaVac or BNT162b2 vaccine following two-dose CoronaVac vaccine regimen. J Med Virol. 2022 Jan;94(1):39-41. doi: 10.1002/jmv.27350. Epub 2021 Sep 21. PMID: 34536028.

-Costa Clemens SA, Weckx L, Clemens R, Almeida Mendes AV, Ramos Souza A, Silveira MBV, da Guarda SNF, de Nobrega MM, de Moraes Pinto MI, Gonzalez IGS, Salvador N, Franco MM, de Avila Mendonça RN, Queiroz Oliveira IS, de Freitas Souza BS, Fraga M, Aley P, Bibi S, Cantrell L, Dejnirattisai W, Liu X, Mongkolsapaya J, Supasa P, Screaton GR, Lambe T, Voysey M, Pollard AJ; RHH-001 study team. Heterologous versus homologous COVID-19 booster vaccination in previous recipients of two doses of CoronaVac COVID-19 vaccine in Brazil (RHH-001): a phase 4, non-inferiority, single blind, randomised study. Lancet. 2022 Feb 5;399(10324):521-529. doi: 10.1016/S0140-6736(22)00094-0. Epub 2022 Jan 21. PMID: 35074136; PMCID: PMC8782575.

-ÇaÄŸlayan D, Süner AF, Åžiyve N, Güzel I, Irmak Ç, IÅŸik E, Appak Ö, Çelik M, Öztürk G, Alp ÇavuÅŸ S, Ergör G, Sayiner A, Ergör A, Demiral Y, Kiliç B. An analysis of antibody response following the second dose of CoronaVac and humoral response after booster dose with BNT162b2 or CoronaVac among healthcare workers in Turkey. J Med Virol. 2022 Jan 25. doi: 10.1002/jmv.27620. Epub ahead of print. PMID: 35075655.

Major concerns:

The results are well analysed but some relevant data are missing. It is important to include the adverse events associated with the booster to support the safety of the heterologous combination. Additionally, the comparison with a group of people received an homologous BBIBP-CorV boost is highly desirable to demonstrate the superiority of the heterologous regimen.

Author Response

Reviewer 2 and Responses:

  1. The manuscript “Evaluation of the humoral immune response of a heterologous vaccination between BBIBP-CorV and BNT162b2 with a temporal separation of 7 months, in Peruvian healthcare workers with and without a history of SARS-CoV-2 infection” submitted by Hueda-Zavaleta M. et al. evaluates the humoral immunity elicited by heterologous combination of two different SARS-CoV-2 vaccine platforms, the BBIBP-CorV vaccine based on inactivated virus and the BNT162b2 vaccine based on mRNA. The authors analysed the longevity of humoral response (anti-RBD binding IgG antibodies and NAbs) induced by two BBIBP-CorV immunizations at 6 months after the last dose and the impact of a late boost with the heterologous BNT162b2 vaccine on the previous immunity. A robust increase of humoral immunity was detected after the heterologous mRNA boost, supporting previous findings.
  • Thank you very much for your consideration.
  1. The authors claimed that they could not find any study that evaluate the heterologous booster of BNT162b2 after two doses with BBIBP-CorV to highlight the novelty of the study, but several previous studies have addressed this evaluation:
  • The reviewer is right, and we appreciate the comment. We have modified the focus of the study; we have also substantially changed the discussion of the study, adding part of the research that you cited and following the suggestions of the other reviewers.
  1. The results are well analysed but some relevant data are missing. It is important to include the adverse events associated with the booster to support the safety of the heterologous combination. Additionally, the comparison with a group of people received a homologous BBIBP-CorV boost is highly desirable to demonstrate the superiority of the heterologous regimen.
  • - We appreciate the observation and suggestion of the reviewer, unfortunately in the designed protocol, this variable (adverse events) was not considered, and we could not report it.
  • - On the other hand, the use of the BBIBP-CorV vaccine as a homologous booster was not approved in Peru; only the heterologous booster BNT162b2 was approved, so we only have this information.
  • - We will add both observations in our limitations.

Round 2

Reviewer 2 Report

The amendments made by the authors have helped considerably.  The discussion in particular is much improved.  Sections of the text can be further improved by a native English speaker, but these are relatively few in number.

Reviewer 3 Report

The manuscript has been extensively modified according to the reviewer suggestions. It can be published in the present form.